Digging deeper: new gene order rearrangements and distinct patterns of codons usage in mitochondrial genomes among shrimps from the Axiidea, Gebiidea and Caridea (Crustacea: Decapoda)

Tan Mun Hua 1 2
Gan Han Ming 1 2 gan.han.ming@monash.edu
Lee Yin Peng 1 2
Poore Gary C.B. 3
http://orcid.org/0000-0003-1848-6267 Austin Christopher M. 1 2 4
1 School of Science, Monash University Malaysia , Bandar Sunway, Selangor , Malaysia
2 Genomics Facility, Tropical Medicine and Biology Platform, Monash University Malaysia , Bandar Sunway, Selangor , Malaysia
3 Museums Victoria , Melbourne, VIC , Australia
4 School of Life and Environmental Sciences, Deakin University , Burwood, VIC , Australia
Huang Xiaolei
Electronic publication date: 2017 Mar 1
Publication date: 2017
Volume: 5
Electronic Location ID: e2982
Received 2016 Aug 11; Accepted 2017 Jan 12
Copyright: © 2017 Tan et al.
Copyright year: 2017
Copyright holder: Tan et al.
License: This is an open access article distributed under the terms of the Creative Commons Attribution License, which permits unrestricted use, distribution, reproduction and adaptation in any medium and for any purpose provided that it is properly attributed. For attribution, the original author(s), title, publication source (PeerJ) and either DOI or URL of the article must be cited.
License URL: https://creativecommons.org/licenses/by/4.0/

Keywords: True shrimps, Mitochondrial genomics, Codon usage, Ghost shrimps, Gene order, Phylogenetics, Shrimps

Funding: Monash University Malaysia Tropical Medicine and Biology Multidisciplinary Platform Funding for this work was provided by the Monash University Malaysia Tropical Medicine and Biology Multidisciplinary Platform. The funders had no role in study design, data collection and analysis, decision to publish or preparation of the manuscript.

==============================
Background

Whole mitochondrial DNA is being increasingly utilized for comparative genomic and phylogenetic studies at deep and shallow evolutionary levels for a range of taxonomic groups. Although mitogenome sequences are deposited at an increasing rate into public databases, their taxonomic representation is unequal across major taxonomic groups. In the case of decapod crustaceans, several infraorders, including Axiidea (ghost shrimps, sponge shrimps, and mud lobsters) and Caridea (true shrimps) are still under-represented, limiting comprehensive phylogenetic studies that utilize mitogenomic information.

Methods

Sequence reads from partial genome scans were generated using the Illumina MiSeq platform and mitogenome sequences were assembled from these low coverage reads. In addition to examining phylogenetic relationships within the three infraorders, Axiidea, Gebiidea, and Caridea, we also investigated the diversity and frequency of codon usage bias and mitogenome gene order rearrangements.

Results

We present new mitogenome sequences for five shrimp species from Australia that includes two ghost shrimps, Callianassa ceramica and Trypaea australiensis, along with three caridean shrimps, Macrobrachium bullatum, Alpheus lobidens, and Caridina cf. nilotica. Strong differences in codon usage were discovered among the three infraorders and significant gene order rearrangements were observed. While the gene order rearrangements are congruent with the inferred phylogenetic relationships and consistent with taxonomic classification, they are unevenly distributed within and among the three infraorders.

Discussion

Our findings suggest potential for mitogenome rearrangements to be useful phylogenetic markers for decapod crustaceans and at the same time raise important questions concerning the drivers of mitogenome evolution in different decapod crustacean lineages.

Introduction

Sequencing of animal mitochondrial genomes has exploded in recent years. Over 40,000 animal mitogenomes are currently lodged on the NCBI database compared to fewer than 5,000 sequences a decade ago. As mitogenomic data have accumulated, it has become apparent that mitochondrial gene order is not as conserved as first thought, and that interesting and phylogenetically useful patterns that invite further research are emerging (Boore, 2006; Dowton, Castro & Austin, 2002; Gan et al., 2016; Gissi, Iannelli & Pesole, 2008; Lin et al., 2012; Poulsen et al., 2013; Tan et al., 2015).

In addition to gene order evolution, properties of mitogenome sequences that are of phylogenetic significance have emerged that are of interest to comparative mitogenomics such as gene loss and duplications, AT bias, strand asymmetry in nucleotide composition, length, and structure of the control region, features of intergenic non-coding regions, codon usage, variation in gene length, variation in start and stop codons, gene diversity levels, mutation rates, and signals of selection and secondary structures of ribosomal genes (Bauzà-Ribot et al., 2009; Gissi, Iannelli & Pesole, 2008; Jia & Higgs, 2008; Li, Huang & Lei, 2015; Oliveira et al., 2008; Poulsen et al., 2013; Qian et al., 2011; Shoemaker et al., 2004). However, despite the rapidly accumulating mitogenomic resources, there are gaps in taxonomic representation and more data are required to fully evaluate the usefulness of mitogenome gene rearrangements as phylogenetic markers in specific groups (Mao, Gibson & Dowton, 2014; Tan et al., 2015) and for broadly based comparative studies to detect patterns and investigate evolutionary hypotheses (Boore, 2006; Castellana, Vicario & Saccone, 2011; Gissi, Iannelli & Pesole, 2008; Jiang et al., 2007).

In general, the overall structure and function of animal mitochondrial genomes are remarkably stable. It is a circular, double-stranded DNA molecule of usually 15–20 kb in length, generally containing a consistent set of 37 genes made up of two ribosomal RNA genes (rRNA), 13 protein-coding genes (PCG) and 22 transfer RNA genes (tRNA) (Bernt et al., 2013; Castellana, Vicario & Saccone, 2011; Gissi, Iannelli & Pesole, 2008). Intergenic regions are usually minimal, although all species contain at least one large AT-rich region associated with strand replication e.g., the control region. Mutations in the mitochondrial DNA can range from point mutations and infrequent insertions/deletions to gene order rearrangements. However, the type, scale and distribution of mutations can vary widely across various taxonomic levels (Gissi, Iannelli & Pesole, 2008; Tan et al., 2015).

The most common mitogenome gene order rearrangements involve the translocation of single tRNA genes and occasionally with a change of transcriptional polarity or duplication. Less frequent is the repositioning of multiple tRNAs, duplication of the control region or changes to the order and orientation of protein coding and rRNA genes. The paradox of mitogenome gene rearrangements is that the molecule can be highly conserved among phylogenetically distant species such as some insect and decapod crustacean species but can, in a restricted number of taxonomic groups, also vary substantially among species in the same family or genus. Examples of taxa with a concentration of reported mitogenome gene rearrangements include ticks (Fahrein et al., 2007; Shao et al., 2005), hymenopterans (Dowton et al., 2003), gulper eels (Poulsen et al., 2013), salamanders (Chong & Mueller, 2013), tunicates (Gissi, Iannelli & Pesole, 2004), and in several crustacean groups (Kilpert & Podsiadlowski, 2006; Kim et al., 2012; Miller et al., 2004; Stokkan et al., 2016; Tan et al., 2015).

Codon usage is another aspect of comparative mitogenomics that is attracting increasing attention as research moves from describing patterns to understanding them within an evolutionary and molecular genetic context (Castellana, Vicario & Saccone, 2011; Gissi, Iannelli & Pesole, 2008; Jiang et al., 2007). Amino acids can be encoded by two to six codons, but alternative codons for the same amino acid often do not occur at equal frequencies either between species for the same gene or between different genes in the one species. Patterns of differential codon usage have been attributable to selection, variable mutation rates, translational efficiency, and random factors (genetic drift) (Castellana, Vicario & Saccone, 2011; Jia & Higgs, 2008; Prat et al., 2009; Whittle & Extavour, 2015). However, it has been rarely addressed among crustacean species (Cook, Yue & Akam, 2005; García-Machado et al., 1999; Rota-Stabelli et al., 2013).

As in other major animal groups, mitogenome sequences are becoming increasingly available for decapod crustacean species, contributing to the understanding of the evolution of this taxonomically challenging group due to its high diversity, deep lineages, and highly flexible body plan (Shen, Braband & Scholtz, 2013; Tan et al., 2015). In addition, intriguing and taxonomically unevenly distributed gene order rearrangements are emerging, requiring further investigation and raising questions regarding the dynamics and drivers of mitogenome gene order evolution in several groups (Gan et al., 2016b; Kilpert & Podsiadlowski, 2006; Miller et al., 2004). Two of the less well-represented decapod crustacean infraorders in mitochondrial databases are the shrimp infraorders Axiidea (ghost shrimps, sponge shrimps, and mud lobsters) and Caridea (true shrimps) (Lin et al., 2012; Tan et al., 2015). Further, the phylogenetic relationships within and among these shrimp groups remain largely unresolved and disputed (Lin et al., 2012; Timm & Bracken-Grissom, 2015). A major limiting factor to the resolution of phylogenetic relationships within and among these shrimp lineages and the determination of the distribution and evolutionary significance of mitogenome gene order rearrangements is inadequate taxon sampling (Lin et al., 2012; Shen, Braband & Scholtz, 2013; Tan et al., 2015).

At the present time, the Axiidea, Gebiidea, and Caridea are represented by only six, five and 17 complete mitogenomes respectively on the NCBI public database. To support ongoing phylogenetic and comparative mitogenomic studies, this paper reports five new mitogenome sequences of shrimp species sampled from Australia. These include two mitogenomes from the ghost shrimp, Callianassa ceramica Fulton & Grant, 1906 and Trypaea australiensis Dana, 1852 (Callianassidae), and three from caridean shrimps Macrobrachium bullatum Fincham, 1987 (Paleamonidae), Alpheus lobidens De Haan, 1849 (Alpheidae) and Caridina cf. nilotica Roux, 1833 (Atyidae), each of which represents highly diverse superfamilies, families and genera within Caridea. This study compares the mitogenomic features of these five species together with additional representatives of their infraorders and Gebiidea available from the NCBI database. In addition to exploring evolutionary relationships within each infraorder, we uncover distinctive signatures and patterns with respect to sequence composition, codon usage bias, and gene rearrangements that can possibly act as synapomorphies for specific shrimp taxonomic groups, suggesting the potential of these features for phylogenetic inferences at different evolutionary scales.

Materials and Methods

Sample collection

Two species belonging to the infraorder Axiidea (C. ceramica, T. australiensis) and three from Caridea (Macrobrachium bullatum, A. lobidens, Caridina cf. nilotica) were collected from different locations in Australia (Table 1). C. ceramica is represented by two individuals: one from a vouchered specimen (Museum Victoria J40715; GenBank accession number KU350630.1); and the other a previously sequenced sample published incorrectly as T. australiensis (KM501040.1) (Gan et al., 2016). The mitogenome sequence for this latter sample is now registered as C. ceramica under the accession number KU726823.1 and has a genetic similarity of 99.8% to the COI region of a vouchered C. ceramica specimen (Museum Victoria J70519) collected from the same general locality.

Table 1 List of source and accession number of mitogenome samples.

	Callianassa ceramica	Callianassa ceramica	Trypaea australiensis	Alpheus lobidens	Caridina cf. nilotica	Macrobrachium bullatum	
Family	Callianassidae	Callianassidae	Callianassidae	Alpheidae	Atyidae	Palaemonidae	
Subfamily	Callianassinae	Callianassinae	Callianassinae	N/A	N/A	Palaemoninae	
Location	Anglesea, South west of Geelong, Victoria	South of Port Authority Pier, Queenscliff, Victoria	Stony Point, Western Port Bay, Victoria	Nightcliff, Darwin, Northern Territory	Amy Ward Drive, Darwin, Northern Territory	Rapid Creek, Darwin, Northern Territory	
Voucher	N/A	NMV J40715	NMV J40711	MAGNT Cr.18581	N/A	N/A	
Verification	99.8% (660 bp COI)1	N/A	N/A	N/A	100% (326 bp 16S)2	100% (447 bp 16S)3	
Accession #	KU362925.1	KU350630.1	KM501040.2	KP276147.1	KU726823.1	KM978918.1	
Notes:

NMV, Museum Victoria; MAGNT, Museum and Art Galleries of the Northern Territory.

1 NMV J70519, Point Roadknight, Anglesea, Victoria.

2 Page, von Rintelen & Hughes (2007) – DQ478508.1 – MAGNT Cr. 9399.

3 Murphy & Austin (2004) – AY282778.1.

To maintain continuity with the original NCBI accession number and species name, the original mitogenome sequence lodged on NCBI for T. australiensis was updated with the newly sequenced mitogenome from a vouchered T. australiensis specimen (Museum Victoria J40711; GenBank accession number: KM501040.2). The accession numbers for each species and associated collecting and identification-related information are detailed in Table 1 and Data S3, including voucher numbers for specimens lodged in Museum Victoria, Melbourne (NMV) and the Museum and Art Gallery of the Northern Territory, Darwin (MAGNT).

Next-generation sequencing and mitogenome assembly

Purification of ethanol-preserved tissue and partial whole genome sequencing (2 × 75 bp for T. australiensis and 2 × 250 bp for others) was performed on the Illumina MiSeq platform as previously described (Gan et al., 2016), after which each mitogenome was assembled with IDBA_UD v.1.1.1 (Peng et al., 2012) and annotated using MITOS (Bernt et al., 2013). Circular mitogenome maps were drawn with BRIG v.0.9.5 (Alikhan et al., 2011). Summary statistics including gene boundaries and length, strand, nucleotide composition, intergenic nucleotides, and number of genes were compiled with MitoPhAST v.1.0 (Tan et al., 2015). Alignment of whole mitogenome sequences and calculation of pair-wise nucleotide identities were performed with SDT v.1.2 (Muhire, Varsani & Martin, 2014).

Gene order analysis

Along with the five mitogenomes sequenced in this study, sequences from 28 other complete mitogenomes from the three infraorders were obtained from NCBI’s RefSeq database (Table 2) for comparative analyses. Arrangements of genes for each of these 33 mitogenomes were compared with all other existing decapod mitogenomes in RefSeq to identify potential novel gene orders unreported by previous studies. Mitogenomes that exhibit gene orders differing from that of the pancrustacean ground pattern (Boore, Lavrov & Brown, 1998) were re-annotated with MITOS (Bernt et al., 2013) to confirm that differences observed are not due to misannotations. Any observed misannotations (e.g., missing genes, incorrect gene boundaries) were corrected before proceeding to further comparative and phylogenetic analyses.

Table 2 List of samples and their corresponding accession numbers included in phylogenetic and comparative analyses.

Infraorder	Family	Species	Accession	Reference	
Axiidea	Callianassidae	Callianassa ceramica	KU350630.1	This study	
		Callianassa ceramica	KU362925.1	Gan et al. (2016)	
		Corallianassa coutierei	NC_020025.1	Shen, Braband & Scholtz (2013)	
		Nihonotrypaea japonica	NC_020351.1	Kim et al. (2013)	
		Nihonotrypaea thermophila	NC_019610.1	Lin et al. (2012)	
		Paraglypturus tonganus	NC_024651.1	Kim et al. (2016b)	
		Trypaea australiensis1	KM501040.2	This study	
	Strahlaxiidae	Neaxius acanthus2	NC_019609.1	Lin et al. (2012)	
Gebiidea	Thalassinidae	Thalassina kelanang	NC_019608.1	Lin et al. (2012)	
	Upogebiidae	Austinogebia edulis	NC_019606.1	Lin et al. (2012)	
		Upogebia major	NC_019607.1	Lin et al. (2012)	
		Upogebia pusilla	NC_020023.1	Shen, Braband & Scholtz (2013)	
		Upogebia yokoyai	NC_025943.1	Yang et al. (2016)	
Caridea	Alvinocarididae	Alvinocaris chelys	NC_018778.1	Yang et al. (2012)	
		Alvinocaris longirostris	NC_020313.1	Yang et al. (2013)	
		Nautilocaris saintlaurentae	NC_021971.1	Kim, Pak & Ju (2015a)	
		Opaepele loihi	NC_020311.1	Yang et al. (2013)	
		Rimicaris exoculata	NC_027116.1	Yu et al. (2015)	
		Rimicaris kairei	NC_020310.1	Yang et al. (2013)	
	Alpheidae	Alpheus distinguendus	NC_014883.1	Qian et al. (2011)	
		Alpheus lobidens	KP276147.1	This study	
	Atyidae	Caridina gracilipes	NC_024751.1	Xu et al. (2016)	
		Caridina cf. nilotica	KU726823.1	This study	
		Halocaridina rubra	NC_008413.1	Ivey & Santos (2007)	
		Neocaridina denticulata	NC_023823.1	Yu, Yang & Yang (2014)	
		Paratya australiensis	NC_027603.1	Gan et al. (2016b)	
	Palaemonidae	Macrobrachium bullatum	KM978918.1	This study	
		Macrobrachium lanchesteri	NC_012217.1	L. Ngernsiri & P. Sangthong, 2016, unpublished data	
		Macrobrachium nipponense	NC_015073.1	Ma et al. (2011)	
		Macrobrachium rosenbergii	NC_006880.1	Miller et al. (2005)	
		Palaemon carinicauda	NC_012566.1	Shen et al. (2009)	
		Palaemon gravieri	NC_029240.1	Kim et al. (2015b)	
		Palaemon serenus	NC_027601.1	Gan et al. (2016b)	
Dendrobranchiata (outgroup)	Sergestidae	Acetes chinensis	NC_017600.1	Kim et al. (2012)	
	Penaeidae	Farfantepenaeus californiensis	NC_012738.1	Gutiérrez-Millán et al. (2002)	
		Fenneropenaeus chinensis	NC_009679.1	Shen et al. (2007)	
		Fenneropenaeus merguiensis	NC_026884.1	Zhang et al. (2016a)	
		Fenneropenaeus penicillatus	NC_026885.1	Zhang et al. (2015)	
		Litopenaeus vannamei	NC_009626.1	L. Ngernsiri & P. Sangthong, 2016, unpublished data	
		Marsupenaeus japonicas	NC_007010.1	Yamauchi et al. (2004)	
		Metapenaeopsis dalei	NC_029457.1	Kim et al. (2016a)	
		Metapenaeus ensis	NC_026834.1	Zhang et al. (2016b)	
		Parapenaeopsis hardwickii	NC_030277.1	Mao et al. (2016)	
		Penaeus monodon	NC_002184.1	Wilson et al. (2000)	
	Solenoceridae	Solenocera crassicornis	NC_030280.1	Y. Yuan et al., 2016, unpublished data	
Note:

1 Mitogenome from taxonomically verified T. australiensis sample resubmitted as version two under same accession number.

2 Neaxius acanthus from Taiwan was wrongly identified as Neaxius glyptocercus by Lin et al. (2012).

Phylogenetic analysis

Mitogenomes listed in Table 2 were subject to phylogenetic analysis to establish the evolutionary relationships of species within each of the infraorders to provide a framework for interpreting mitogenome gene rearrangements. MitoPhAST v.1.0 (Tan et al., 2015) was used to extract individual PCG amino acid sequences, and these protein sequences were then separately aligned with MAFFT v.7.222 (Katoh & Standley, 2013), followed by trimming with trimAl v.1.4 (Capella-Gutiérrez, Silla-Martínez & Gabaldón, 2009). For nucleotide level analyses, PCG nucleotide sequences were manually extracted and fed to TranslatorX v.1.1 (Abascal, Zardoya & Telford, 2010), which aligns nucleotide sequences guided by amino acid translations and then trimmed with Gblocks v.0.19b (Castresana, 2000). On the other hand, rRNA was aligned with MAFFT v.7.222 (mafft-linsi) (Katoh & Standley, 2013) and trimmed with trimAl v.1.4 (Capella-Gutiérrez, Silla-Martínez & Gabaldón, 2009). Finally, mitochondrial PCG and rRNA sequences were concatenated into super-alignments to make up the following datasets: 13 PCG (aa) [3,591 characters]

13 PCG (nt) [9,642 characters]

13 PCG (aa) + 12S rRNA + 16S rRNA [5,694 characters]

13 PCG (nt) + 12S rRNA + 16S rRNA [11,755 characters]

Maximum-likelihood (ML) tree inference with ultrafast bootstrap (UFBoot) branch supports (Minh, Nguyen & von Haeseler, 2013) was performed using IQ-TREE v.1.5.0 (Nguyen et al., 2015), which also implements model selection to find the best-fit partitioning scheme. Super-alignments for all datasets were partitioned based on genes. An additional analysis for Dataset B that further partitions it according to first, second and third codon positions was also performed. For Bayesian inference, the same super-alignments generated from all datasets were analysed using Exabayes v.1.4.2 (Aberer, Kobert & Stamatakis, 2014). For each analysis, four independent runs were carried out concurrently for five million iterations each with 25% of initial samples discarded as burn-in. Convergence of chains was checked by ensuring the average standard deviation of split frequencies (asdsf) is below 0.5%, considered to be good convergence according to the Exabayes user guide. Alignments, partitions and best-fit partitioning schemes for all datasets are available as Data S1.

Codon usage analysis

Codon usage (in counts) was calculated using EMBOSS v.6.5.7 (Rice, Longden & Bleasby, 2000) followed by minor adjustments based on the Invertebrate Mitochondrial Code (genetic code = 5). Comparisons among the three lineages of shrimps were made by applying the chi-square test to the pooled codon usage counts for species from each infraorder. Relative synonymous codon usage (RSCU) values were calculated by taking the ratio of the number of times a codon appears to the expected frequency of the codon if all synonymous codons for a same amino acid are used equally (Sharp & Li, 1987). Patterns of variation among individuals in RSCU values were summarized using multidimensional scaling (MDS) based on Euclidean dissimilarities implemented in XLSTAT v.2015.4.01.20978 (Addinsoft, 2010).

Results

Mitogenome composition

Mitogenomes for specimen J40715 of C. ceramica (16,899 bp, 130× cov), T. australiensis (15,485 bp, 86× cov), M. bullatum (15,774 bp, 27× cov), A. lobidens (15,735 bp, 60× cov) and Caridina cf. nilotica (15,497 bp, 63× cov) were assembled into complete circular sequences, annotated (Fig. 1), and deposited in GenBank with accession numbers listed in Table 1. Four of the mitogenomes contain the typical 13 PCG, two ribosomal RNA genes, 22 transfer RNA genes and one long non-coding region while A. lobidens has an additional trnQ flanked by the ND4L and trnT genes (Fig. 1). Detailed composition of each mitogenome can be found in Table S1 while general information on % AT and lengths for all mitogenomes included in this study are in Table S2. The mitogenomes are AT rich (58.9–69.7%), with A. lobidens having the lowest AT content, matching closely to Alpheus distinguendus (60.2%), the only other species of Alpheus having a published mitogenome sequence. Gene lengths are typical but Callianassa and Trypaea have an elevated proportion of intergenic nucleotides, with some spacers in the order of 200 bp in length. This is significantly larger than for other members of the Axiidea, but similar to spacers reported for the Gebiidea. C. ceramica (KU350630.1) has an unusually long control region of 2,036 bp, whereas for all the other taxa it is less than 1,000 bp, including the closely-related T. australiensis (587 bp). However, the control region for the conspecific C. ceramica (KU362925.1) is very similar (1,978 bp) and these two specimens are also very similar in terms of % AT (69.7 and 70.2%). A matrix of pair-wise identities of whole mitogenome sequence alignments can be found in Fig. S1.

Figure 1 Circular representation of three caridean and two axiidean species.

These figures show composition and order of protein-coding genes (blue), ribosomal RNAs (orange), transfer RNAs (purple) and large non-coding region (grey) for the following mitogenomes: (A). Alpheus lobidens, (B). Caridina cf. nilotica, (C). Macrobrachium bullatum, (D). C. ceramica, (E). Trypaea australiensis.

Manual inspection and MITOS annotation identified multiple erroneously annotated crustacean mitogenomes in GenBank RefSeq database

The re-annotated species with gene orders divergent to that of the pancrustacean ground patterns obtained from GenBank’s RefSeq database identified several that require revision. One was found to have a missing protein-coding gene and an extra trnS, and others inverted rRNA and tRNA coordinates and other annotation anomalies as detailed in Table 3. For our study, entries were edited based on MITOS annotations and the revised GenBank files for these entries are included as Data S2.

Table 3 List of samples for which annotations were corrected based on re-annotation with MITOS.

Accession #	Species	Genes involved1	Correction/edits made in this study	
NC_024751.1	Caridina gracilipes	ND2, trnS	Added ND2, removed duplicated trnS	
NC_012217.1	Macrobrachium lanchesteri	rrnS, rrnL	Inverted coordinates for rrnS and rrnL	
NC_020025.1	Corallianassa coutierei	trnI, trnQ	Inverted coordinates for trnI and trnQ	
NC_020351.1	Nihonotrypaea japonica	rrnS, rrnL	Added rrnS and rrnL coordinates	
Note:

1 Genes involved in discrepancies found between NCBI’s RefSeq entry and re-annotation with MITOS.

Whole mitogenomes are consistent with monophyly of infraorders Axiidea, Gebiidea, and Caridea

A total of 33 mitogenome sequences from the three groups of interest were utilized (Axiidea: eight, Gebiidea: five, Caridea: 20) to elucidate phylogenetic relationships, with an additional 12 Dendrobranchiata mitogenomes as an outgroup (Table 2). Trees constructed from every dataset and analysis are available in Fig. S2. All trees place M. bullatum, A. lobidens, and Caridina cf. nilotica as sister taxa to other species from their respective genera with relatively high nodal support (UFBoot ≥ 93%, PP 1.00). These trees also consistently place C. ceramica as sister to T. australiensis (UFBoot 100%, PP 1.00) within Axiidea.

The Bayesian-inferred phylogenetic tree in Fig. 2A, constructed based on amino acid sequences of 13 PCGs and nucleotide sequences of two rRNAs (Dataset C), shows a tree topology that is shared by most trees inferred in this study. Most nodes received maximal support from each analysis and dataset. The greatest levels of uncertainty in terms of phylogenetic placement are mostly relating to the relationships among closely related taxa, such as in the Palaemon (6/10 trees), Rimicaris–Opaepele–Alvinocaris–Nautilocaris (2/10 trees), Upogebia–Austinogebia (1/10 trees), Corallianassa–Paraglypturus (1/10 trees), and Macrobrachium (1/10 trees) clades. The only deeper clade with low support is the placement of Atyidae as the sister clade of Alvinocarididae within Caridea (UFBoot ≥ 90, PP 1.00). Within Axiidea, maximal support is observed for almost all nodes. The most basal split in this infraorder separates Strahlaxiidae (Neaxius acanthus) from Callianassidae. Within Callianassidae, the three major lineages correspond to accepted subfamilies, two represented by one species each and Callianassinae by four species. In as far as it goes, the phylogenetic placements of mitogenome sequences in Axiidea are congruent with the current classification at the family, subfamily, and generic levels (Felder & Robles, 2009) (Fig. 2A). These analyses of just four upogebiid species indicate that Upogebia may be paraphyletic with respect to Austinogebia. Since these two genera are nominally represented by over 120 and eight species respectively, any comment on their status is premature at this stage. The degree of divergence between the species of Rimicaris and Opaepele is small relative to the degree of divergence between congeneric species within Macrobrachium, Alpheus, Caridina, and Alvinocaris.

Figure 2 Phylogenetic relationships and gene order rearrangements within Axiidea, Gebiidea and Caridea.

(A). Phylogenetic tree with support values indicated at each node (top, l-r: ML and BI support for PCG (aa) + 12S + 16S dataset), bottom, l-r: ML and BI support for PCG (nt) + 12S + 16S dataset). Square brackets [ ] around a value indicate that the shown topology is in conflict with that constructed by the specific dataset. If values are absent at a node, maximum support was found for that node based on all phylogenetic inference methods and datasets. Topology shown was inferred from Bayesian analysis of PCG (aa) + 12S + 16S dataset. Codes on branches (Gr, Pa, Ap1, Ap2, Up, Ax1, Ax2) correspond to gene order pattern listed in B while red stars indicate mitogenomes sequenced in this study. (B). Gene order of various groups. Yellow triangles under genes indicate differences in arrangement compared to the ground pancrustacean pattern.

Deviation from the pancrustacean ground pattern is prevalent in the currently sequenced members of the Axiidea and Gebiidea

Most mitogenomes from caridean species have the pancrustacean ground pattern (pattern Gr, Fig. 2B). Those that differ show only minor rearrangements involving the short tRNA genes (patterns Pa, Ap1, and Ap2). In contrast, the mitogenomes of species within Axiidea and Gebiidea exhibit relatively substantial differences in gene order (patterns Up, Ax1, and Ax2) entailing rearrangements of PCG, rRNAs, and a number of tRNAs. An example is pattern Ax2 shown in Fig. 2B, which includes the inversion of the ND1, lrRNA, srRNA, and trnI genes as a block, in addition to the inversion and translocation of trnD from between trnQ and trnM to a position between trnS2 and the putative control region, as well as new placements for trnL and trnV also evident in pattern Ax1.

All gene order novelties relative to the pancrustacean ground pattern are consistent with the relationships depicted by the molecular phylogeny, and in several cases define taxonomic groups at different levels. In Caridea, pattern Pa is common to the three Palaemon species and pattern Ap1 defines the two Alpheus species. Similarly, within Gebiidea, while Thalassina kelanang (family Thalassinidae) has the ground pancrustacean pattern, the other four species are all members of the family Upogebiidae and are united by novel rearrangements involving several tRNA translocations (pattern Up, Fig. 2B). Pattern Ax1, involving the rearrangements of COIII and several tRNAs, is shared among species of Axiidea. The elements of pattern Ax2 that differ from pattern Ax1 support the node joining Callianassa and Trypaea.

Evidence of significant codon usage bias in mitochondrial genomes at the infraorder level

Figure 3 shows there is strong A+T bias in codon usage across the 33 shrimp mitogenomes. RSCU frequencies demonstrate distinct preference for codons with A or T in the third codon position compared to other synonymous codons. Counts of codons used and RSCU values can be found in Table S3. Among the 62 available codons, the four most used codons in all three infraorders are TTT (Phe), TTA (Leu), ATT (Ile), and ATA (Met), all made up of solely A and T nucleotides. Even so, this preference for A+T codons is stronger in Axiidea and Gebiidea mitogenomes and less so in Caridea, most obvious for amino acids Asp (D), His (H), Asn (N), and Tyr (Y). Statistical comparisons show that, for each amino acid, there are significant differences among the three infraorders in the proportions of the codons being used (p-values in Fig. 3). A separate comparison for species with pattern Ax1and Ax2 within the infraorder Axiidea and also with those with the ground pattern (Gr) do not reveal any substantial difference in their codon usage bias (Fig. S3). The MDS plot shows that, for the most part, members of each infraorder cluster together and are largely distinct from the samples from the other infraorders (Fig. 4). A sample of the Gebiidea, T. kelanang, is a maverick, being placed well inside the Caridea cluster and remote from the other members of its infraorder. It also has a very low AT content for this group, being more similar to caridean shrimps. In this context, it is noteworthy that the species is placed as the most basal member of the Gebiidea and is also the only member of the infraorder that has the primitive pancrustacean gene order, which it shares with all other of the members of the Caridea with the exception of some species with minor derived rearrangements (Fig. 2B).

Figure 3 Relative synonymous codon usage values (RSCU) (y-axis) in protein-coding genes of mud shrimps and true shrimps.

Encoded amino acid and its corresponding p-value (> or <0.001) is shown at the top of each box plot while synonymous codons are indicated on the x-axis.

Figure 4 Patterns of variation among individuals based on RSCU values shown using multidimensional scaling (MDS) based on Euclidean dissimilarities.

Discussion

The five new decapod mitogenomes presented in this study considerably expand the number of samples of Axiidea and Caridea currently available for mitogenome-based phylogenetics and comparative mitogenomic studies. Members of the infraorder Gebiidea were also included in this analysis due to its taxonomic history of having been placed with Axiidea in the infraorder Thalassinidea, prior to obtaining recent recognition as its own separate infraorder (Ahyong & O’Meally, 2004; Crandal, Harris & Fetzner, 2000; Robles et al., 2009; Timm & Bracken-Grissom, 2015). The mitogenomic features of these taxa are generally consistent with those for the Decapoda and for the three infraorders (Kim et al., 2013; Lin et al., 2012; Miller et al., 2005) (Table S1). In addition, the high AT content (58.9–73.6%) observed in all of the mitogenomes utilized in this study is typical for the Crustacea and the Arthropoda (Cameron, 2014; Cook, Yue & Akam, 2005; Lin et al., 2012; Shen et al., 2015).

It has been suggested that evolutionary rate and the frequency of rearrangements are independent (Gissi, Iannelli & Pesole, 2008) and this is consistent with results depicted in Fig. 2, which indicate no obvious correlations between the size and distribution of rearrangements and branch lengths (substitution rate) within and between groups. Despite this general observation, some studies have noted an association between higher substitution rates and the occurrence of mitogenome rearrangements involving a transfer of genes between strands (e.g., Stokkan et al., 2016). Although it is expected that codon usage varies between major groups of organisms and among animal phyla (Castellana, Vicario & Saccone, 2011; Gissi, Iannelli & Pesole, 2008; Prat et al., 2009), finding substantial codon usage differences at the infraorder level is somewhat unusual. It is becoming more apparent now that there are divergent patterns in AT content among orders in insects and among the major taxonomic groups in the Malacostraca (see Table 1 in Sun et al. (2009) and Shen et al. (2015)). Studies examining patterns of codon usage in mitogenomes have failed to observe differences at finer taxonomic scales within insects and crustaceans (Cook, Yue & Akam, 2005; Shen et al., 2015; Sun et al., 2009).

Elevated mutational pressure is thought to be a major driver of non-random mitochondrial synonymous variation. However, selection of optimal codons for translational efficiency and genetic drift is also thought to play a role (Castellana, Vicario & Saccone, 2011; Jia & Higgs, 2008; Prat et al., 2009). It is tempting to speculate that the distinctive pattern of codon usage in the Axiidea, and the frequency and extent of mitogenome gene order rearrangements may be correlated with the acquisition of specialized adaptations by members of this infraorder to largely burrowing lifestyles (Lin et al., 2012; Sakai, 2011) compared to the members of the Caridea, which are mostly free living (Bracken, De Grave & Felder, 2009).

The phylogenetic analyses using whole mitogenomes support the monophyly of each of the three infraorders, Axiidea, Gebiidea, and Caridea, with inclusion of more comprehensive taxon sampling than previous studies (Lin et al., 2012; Shen, Braband & Scholtz, 2013; Shen et al., 2015; Tan et al., 2015). Further, the members of Axiidea have a common gene rearrangement that is a potential synapomorphy of the infraorder, and therefore also supports the monophyly of this group. A notable exception to these results is the analysis of Shen et al. (2015), which supported a non-monophyletic Gebiidea, by placing T. kelanang inside a lineage comprising Acheleta, Polychelida, and Caridea. Other studies using a combination of mitochondrial and nuclear gene fragments (Bracken, De Grave & Felder, 2009; Robles et al., 2009) also support the monophyly of these groups, but their relationships with each other and other decapod infraorders have yet to be resolved (Tan et al., 2015). A caveat of our findings and those of other studies is that monophyly cannot be fully tested without comprehensive taxonomic and gene sampling and the inclusion of species from all decapod infraorders (Timm & Bracken-Grissom, 2015).

The contribution of mitogenomic data for more species will be particularly important for exploring and testing evolutionary relationships within Axiidea, given its lack of broad taxonomic representation on the current evolutionary tree. Two of the mitogenomes contributed in this study (C. ceramica, T. australiensis) belong to the family Callianassidae, a diverse group of axiidean shrimps adapted to digging in soft marine sediments. Over 100 species placed in 22 genera and divided between several subfamilies are recognized from this family (Sakai, 2011). However, relationships among the major groups and the definition of subfamily and generic level boundaries are contentious within the Callianassidae (Poore et al., 2014; Sakai, 2011). It is noteworthy that for the Axiidea, the tree-based relationships and the mitogenome rearrangements are entirely consistent with the infraorder, family, subfamily and genus level relationships for the samples included (Fig. 2), and that the phylogenetic tree and the Ax1 rearrangement pattern further supports the monophyly of the Axiidea.

The potential for mitochondrial rearrangement to act as “super” characters for phylogenetic estimation for the arthropod has been recognised by a number of authors (Boore, Lavrov & Brown, 1998; Dowton & Austin, 1999; Dowton, Castro & Austin, 2002). This study further supports this position but also notes that the distribution of rearrangements is uneven across the tree generated in this study and the larger analysis by Tan et al. (2015). Thus, while novel gene order attributes act as useful phylogenetic characters for the Axiidea and Gebiidea, their usefulness appear to be limited for the Caridea, even though the age of the lineages overlap. Similarly, for other major crustaceans groups, lobsters (Astacidea and Achelata), crabs (Brachyura), and penaeid shrimps (Dendrobranchiata), rearrangements are largely absent or minor (Shen et al., 2015; Tan et al., 2015), but freshwater crayfish (Astacidea) and anomuran crabs (Anomura) are phylogenetic “hotspots” for mitogenome gene order evolution in both the number and scale of rearrangements. As pointed out by Mao, Gibson & Dowton (2014), far greater sampling is required to adequately test the phylogenetic utility of observed mitogenome gene rearrangements and identification of suitable models for investigating the evolutionary and molecular drivers that shape the organization and architecture of animal mitogenomes.

In this regard, rapid and efficient methods for mitogenome sequencing using next-generation sequencing (NGS) platforms will accelerate this task (Gan et al., 2016; Timmermans et al., 2016). Unlike more classical methods utilizing long-range polymerase chain reaction (PCR) to amplify the mitogenome into a smaller number of fragments with universal primers, followed by Sanger sequencing (Lin et al., 2012; Miller et al., 2004; Shen, Braband & Scholtz, 2013), the primer-free and shotgun nature of NGS will likely increase the discovery of mitogenome rearrangements as it makes no assumptions about the pre-existing gene order of the species under study. In fact, the availability of a reference whole mitogenome will improve primer design and consequently the success rate of complete mitogenome recovery from members of the same genus (or family) using long-range PCR. It is envisaged with the advent of third generation sequencing technology such as PacBio (Rhoads & Au, 2015) and Nanopore (Branton et al., 2008) sequencing, problematic and repetitive regions commonly associated with the control region can be readily resolved and confirmed.

Lastly, while preparing the dataset for analysis, we identified several misannotated mitogenomes on NCBI (Table 3) although these misannotations were absent in their related publications i.e., the correct gene coordinates and orientations were reported by the authors in their respective publications. We postulate that these discrepancies may have arisen due to errors during the submission of these mitogenomes to public databases. It is also possible, though less likely, that the mitogenomes may have been erroneously edited when they were reviewed by NCBI staff. Hence, this highlights that although there is tremendous gain from having a growing number of mitogenome submissions to public databases as molecular resources, the accuracy of annotations should not be assumed and it is prudent to re-evaluate species with any form of gene order rearrangements or related anomalies before inclusion in datasets for comparative analyses.

Supplemental Information

Supplemental Information 1 Summary of characteristics of mitogenomes presented in this study.

Click here for additional data file.

Supplemental Information 2 Comparison of number of genes, mitogenome lengths, AT content and control region lengths of mitogenomes from Axiidea, Gebiidea and Caridea.

Click here for additional data file.

Supplemental Information 3 Relative synonymous codon usage (RSCU) values from codon usage analysis.

Click here for additional data file.

Supplemental Information 4 Corrected genbank format files for mitogenomes with possible mis-annotations.

Click here for additional data file.

Supplemental Information 5 Corrected genbank format files for mitogenomes with possible mis-annotations.

Click here for additional data file.

Supplemental Information 6 Annotated 5 whole mitogenome reported in this study.

Genbank files containing the annotated whole mitogenome sequence of 5 shrimps reported in this study.

Click here for additional data file.

Supplemental Information 7 Pairwise nucleotide identity of shrimp mitogenomes.

Click here for additional data file.

Supplemental Information 8 Phylogeny of shrimps based on different mitochondrial gene set and sequence type.

All trees were constructed using IQ-TREE with optimized partitioning scheme. Trees were rooted with members of Dendrobranchiata as the outgroup. Red stars next to tip labels indicate mitogenomes reported in this study. PCG, protein coding gene; ML, maximum likelihood; aa, amino acid; nt, nucleotide; BI, Bayesian inference.

Click here for additional data file.

Supplemental Information 9 Codon usage for species with Ax1, Ax2 and Gr mitogenome organization.

Encoded amino acid and its corresponding p-value (> or <0.001) is shown at the top of each box plot while synonymous codons are indicated on the x-axis.

Click here for additional data file.

MHT and HMG would like to acknowledge the Monash University Malaysia High Performance Computing infrastructure for the provision of computational resources. We are also grateful to C. Tudge for collecting and providing the T. australiensis sample and Joanne Taylor, Museum Victoria, for making it available. The authors are also grateful for helpful comments of the anonymous reviews that significantly improved the manuscript.

Additional Information and Declarations

Competing Interests

Author Contributions

Data Deposition

The authors declare that they have no competing interests.

Mun Hua Tan performed the experiments, analysed the data, wrote the paper, prepared figures and/or tables and reviewed drafts of the paper.

Han Ming Gan performed the experiments, analysed the data, wrote the paper and reviewed drafts of the paper.

Yin Peng Lee performed the experiments.

Gary C. B. Poore contributed analysis tools, wrote the paper and reviewed drafts of the paper.

Christopher M. Austin conceived and designed the experiments, analysed the data, contributed reagents/materials/analysis tools, wrote the paper and reviewed drafts of the paper.

The following information was supplied regarding data availability.

The raw data is included in the manuscript as GenBank accession code. The corresponding accession code to each mitogenome reported in this study is listed in Table 1 and also in Data S3. GenBank: KU362925.1; KU350630.1; KM501040.2; KP276147.1; KU726823.1; KM978918.1.

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
