# Peer review of "Digging deeper: new gene order rearrangements and distinct patterns of codons usage in mitochondrial genomes among shrimps from the Axiidea, Gebiidea and Caridea (Crustacea: Decapoda)"

_PeerJ, doi:10.7717/peerj.2982_

## Round 0.1 · original submission · Major Revisions

Besides suggestions on presenting results and certain analyses, there are two points the authors should pay much attention to. One is about the mitochondrial genome sequences used in the present study. The authors should indicate clearly how much new sequences the current paper reports, and address the issue raised by reviewer about Mitochondrial DNA publication. The other is about "comparative mitogenomics", one reviewer argued that the authors need to sample more taxa.

Reviewer 1 ·

Basic reporting

n/a

Experimental design

n/a

Validity of the findings

n/a

Additional comments

KM978918 is now being published in mitochondrial DNA (2015), I am afraid it is not worth to be published.

Full mitochondrial DNA sequencing is now no more new findings and there must be some journals like mitochondrial DNA only depositing mitogenome equences.


Besides mitogenome DNA sequence, major findings in this manuscript is the structural difference in different evolutional relationship.
However, only one (AP1 ) or two (Ap2 or AX2) sequences may not enough to make a conclusion about author's evolutional explanation. From the reason, there must be more sequences from other species belonging to Axiidea and alpheidae.
Without this information, I am afraid there is no useful data for further publication.

Reviewer 2 ·

Basic reporting

For the names of mitochondrial rDNAs in the legend of Figure 2, the alphabet after 12 or 16 should be in upper case "S" rather than lower case "s".

Experimental design

When the authors reviewed the background in the introduction part, they mentioned that "inadequate taxon sampling" had limited the phylogenetic resolving of Decapoda. However, only 3 out of 10 infraorders of Pleocyemata were included as ingroups in this study. And even the word "Pleocyemata" has not been mentioned in the whole manuscript. It could be argued that, with the words "comparative mitogenomics", maybe the authors did not mean "phylogenetic comparisons". But according to Figure 2, it seems that phylogeny is indeed a major frame for the comparisons of gene orders in mitogenomes. Therefore, a phylogeny of Decapoda-Pleocyemata with improved taxon sampling is still necessary in this study. Besides, more data of mitochondrial gene orders of the other infraorders in Pleocyemata should also be considered.

As for the methods of phylogenetic reconstruction, the datasets with only 13 PCGs and the datatype of amino acids should also be analyzed.

Validity of the findings

With the severe absence in taxon sampling, it is quite hard to evaluate the validity of the findings in this study.

Additional comments

For the introduction, it could be better to focus on the points which will be well addressed by this study.

Reviewer 3 ·

Basic reporting

No comments

Experimental design

The experimental design is Ok but the goals are descriptive of a pattern observed rather than hypothesis driven trying to shed light of the causes of the pattern.

Validity of the findings

The results are robust an sound

Additional comments

1) Please state on the main text the coverage of each mitogenomes and also how many reads composed the mitochondrial contig. Please also state the sequencing platform which I guess was Illumina.
2) Please include whether there are nucleotide sequence identity (pair-wise) among the control regions of the different species studied. Please also report sequence identity for other non-coding and other part of the genome.
3) Is codon usage for species with Ax1 arrangement and particularly Ax2 one different to the others? i.e. Do they prefer AT-richer codons (or poor) than other species?
In other words, is there correlation between an increment of gene rearrangements and codon usage pattern?
4) The authors highlight that in mitogenome order Ax2earrangement there is a block including ND1, lrRNA, srRNA but they do not talk that such block does not include the tRNA Val which is translocated between trnG and NAD3. This new placement of trnVal is also is present in Ax1. In pancrsutanean order tRNA Val is located between rrnL (16S) and rrnS (12S).
5) The authors state that there is 64 available codon and that is wrong since this cipher includes two stop codons (TAA-TAG). I think the available codons for protein coding genes in codon table 5 (mitochondrial invertebrate genetic code) is 62.
6) The authors state that gene rearrangements, specially including protein coding genes, are rare- They cite several exception but recently have been published large genome rearrangements mitogenome within a crustacean genus Psudoniphargus (Morten et al 2016 Mitochondrial DNA doi: 10.3109/19401736.2015.1079821)
7) In lane 341, they claim that does not exist correlation between gene rearrangement and genetic distance (higher substitution rates on those species with more gene rearrangements). I do not agree since the species with the mitogenome order Ax1, and particularly Ax2, which block (rrnS, rrnL, nad1, trnD) has moved to positive strand, show longer branch lengths on the tree. See Morten et al 2016 for further explanation.
9) Please include on Fig2 the substitution nucleotide rate for each branch.
10) Finally, I suggest to use partition finder to estimate the best partitioning scheme for the 13 protein coding genes. Generally, 3 partitions by codons (1st, 2nd, 3rd) or 6 (1st pos, 1st neg, 2nd pos, 2nd neg, 3pos, 3neg) is fitting better mitochondrial protein sequences rather than by gene (http://www.robertlanfear.com/partitionfinder/).
The use an incorrect substitution model generally retrieve a wrong tree topology

---

## Round 0.2 · Minor Revisions

Considering that one reviewer questioned the usage of "comparative mitogenomics" in the title, I suggest the authors make a slight change of your title. Then I will accept this submission. For example, you may use "Digging deeper: new gene order rearrangements and distinct patterns of codon usage in mitochondrial genomes among shrimps from the Axiidea, Gebiidea and Caridea (Crustacea: Decapoda)".

Reviewer 1 ·

Basic reporting

All the issues I raised were solved properly including publishing data. They introduced 5 mitogenome sequences with proposed evolutional relationship. Although I am still not convincing that introduction of mitogenome sequences is still a topic for this journal, their data may be worth to published considering relatively low mitogenome numbers beloning to decadpod species.

Experimental design

overall experimental design was typical and there is no controversial issue

Validity of the findings

As I mentioned, their data may be worth to published considering relatively low mitogenome numbers beloning to decadpod species.

Additional comments

All the issues I raised were solved properly including publishing data. They introduced 5 mitogenome sequences with proposed evolutional relationship. Although I am still not convincing that introduction of mitogenome sequences is still a topic for this journal, their data may be worth to published considering relatively low mitogenome numbers beloning to decadpod species.

Reviewer 2 ·

Basic reporting

No comments

Experimental design

No comments

Validity of the findings

No comments

Additional comments

The taxon sampling of this study is quite incomplete, which makes the so-called comparative mitogenomics do not make sense.

Reviewer 3 ·

Basic reporting

The revised version of the manuscript corrected all drawbacks and minor errors denoted by the 3 reviewers so I now can recommend its publication in the present form.

Experimental design

Correct.

Validity of the findings

The finding are valid but they are interesting for people working on very similar topics only.

Additional comments

None

---

## Round 0.3 · accepted · Accept

The authors have made appropriate revisions based on the review comments. Considering relatively few mitochondrial genome sequences of Decapoda species have been reported and the submission provides some new information, I would like to accept it now.